# Causal Association between Circulating Metabolites and Dementia: A Mendelian Randomization Study

**DOI:** 10.3390/nu16172879

**Published:** 2024-08-28

**Authors:** Hong-Min Li, Cheng-Shen Qiu, Li-Ying Du, Xu-Lian Tang, Dan-Qing Liao, Zhi-Yuan Xiong, Shu-Min Lai, Hong-Xuan Huang, Ling Kuang, Bing-Yun Zhang, Zhi-Hao Li

**Affiliations:** Department of Epidemiology, School of Public Health, Southern Medical University, Guangzhou 510515, China; hongminli2209@163.com (H.-M.L.); qiuchsh2021@gmail.com (C.-S.Q.); duliying02@163.com (L.-Y.D.); tangxulianpm@163.com (X.-L.T.); liaodq0926@163.com (D.-Q.L.); xiongzhy6@foxmail.com (Z.-Y.X.); laishumin0119@163.com (S.-M.L.); huanghongxuan0113@163.com (H.-X.H.); kuangling22@163.com (L.K.); zhangbingyun00@163.com (B.-Y.Z.)

**Keywords:** metabolites, dementia, Alzheimer’s disease, vascular dementia, Mendelian randomization

## Abstract

The causal association of circulating metabolites with dementia remains uncertain. We assessed the causal association of circulating metabolites with dementia utilizing Mendelian randomization (MR) methods. We performed univariable MR analysis to evaluate the associations of 486 metabolites with dementia, Alzheimer’s disease (AD), and vascular dementia (VaD) risk. For secondary validation, we replicated the analyses using an additional dataset with 123 metabolites. We observed 118 metabolites relevant to the risk of dementia, 59 of which were lipids, supporting the crucial role of lipids in dementia pathogenesis. After Bonferroni adjustment, we identified nine traits of HDL particles as potential causal mediators of dementia. Regarding dementia subtypes, protective effects were observed for epiandrosterone sulfate on AD (OR = 0.60, 95% CI: 0.48–0.75) and glycoproteins on VaD (OR = 0.89, 95% CI: 0.83–0.95). Bayesian model averaging MR (MR-BMA) analysis was further conducted to prioritize the predominant metabolites for dementia risk, which highlighted the mean diameter of HDL particles and the concentration of very large HDL particles as the predominant protective factors against dementia. Moreover, pathway analysis identified 17 significant and 2 shared metabolic pathways. These findings provide support for the identification of promising predictive biomarkers and therapeutic targets for dementia.

## 1. Introduction

Dementia, in particular Alzheimer’s disease (AD) and vascular dementia (VaD), is the major cause of disability among older individuals globally, resulting in serious physical, financial and social consequences [1]. The prevalence of dementia was estimated to exceed 50 million individuals globally in 2019, with projections indicating more than a twofold increase by 2050 [2]. Due to the absence of well-established disease-modifying therapies, it is an urgent public health priority to explore the potential biomarkers for early identification and prevention of dementia [3].

Metabolomics thoroughly quantifies small molecular metabolites in specific tissues or biofluids, reflecting genetic, environmental, and pathological changes associated with disease progression [4]. Recently, converging evidence has indicated that metabolic dysfunction may be a major hallmark and cause of dementia [5]. Previous observational studies have revealed various circulating metabolites, such as amino acids, fatty acids, and lipids, related to dementia [6,7], but further studies have provided contradictory findings [8,9]. The inconsistencies may arise due to variations in sample characteristics and study designs. Additionally, even with rigorous adjustments for potential confounders, observational studies are still vulnerable to residual unmeasured confounding factors. Therefore, evidence solely from observational studies is insufficient to clarify the causal relationships of circulating metabolites with dementia risk for early identification and prevention.

Mendelian randomization (MR) is an emerging epidemiological method utilizing genetic instrumental variables (IVs) to examine the causal relationship of exposures with outcomes, thus overcoming the inevitable defects of conventional observational studies including reverse causation and confounding [10]. Recent technological advances in mass spectrometry and genotyping have enabled genome-wide association studies (GWASs) to comprehensively reveal the genetic determinants of hundreds of circulating metabolites [11]. In this context, MR design provides a new strategy for accessing the causal association of circulating metabolites with dementia by integrating genomics and metabolomics. However, no large-scale MR analysis has been conducted to systematically estimate the association between circulating metabolites and dementia.

Therefore, we utilized systematic MR analysis to (1) comprehensively evaluate the causal effects of circulating metabolites on dementia; (2) disentangle the prioritization of the predominant metabolites for dementia risk; and (3) identify potential metabolic pathways to enhance the comprehension of the underlying mechanism of dementia.

## 2. Materials and Methods

### 2.1. Study Design

An outline of our study design is presented in Figure 1. Initially, we performed univariable MR analysis to evaluate the causal relationships between circulating metabolites and the risk of outcomes. Subsequently, we performed Bayesian model averaging MR (MR-BMA) analysis to prioritize the metabolites that contribute to the risk of outcomes. Furthermore, we conducted a metabolic pathway analysis to explore the underlying mechanism of dementia. Our analysis followed the Strengthening the Reporting of Observational Studies in Epidemiology using Mendelian Randomization (STROBE-MR) guidelines [12].

### 2.2. GWAS Data Sources for Circulating Metabolites and Dementia

The summary-level datasets for circulating metabolites were obtained from two large-scale comprehensive metabolite GWASs. Shin et al. investigated 486 circulating metabolites in 7824 European adults with approximately 2.1 million SNPs from 2 cohorts [11]. Of the 486 metabolites, 309 known metabolites could be further assigned to eight broad metabolic groups (amino acids, carbohydrates, cofactors and vitamins, energy, lipids, nucleotides, peptides, and xenobiotic metabolism). Moreover, Kettunen et al. investigated 123 circulating metabolites, including lipoprotein lipids and lipid subclasses, fatty acids, amino acids, and glycolysis precursors, in 24,925 European individuals with approximately 12 million SNPs from 14 cohorts [13]. The lipoprotein subclass-specific lipids have supplemented the corresponding deficiencies of previous GWAS summary statistics.

The GWAS datasets for dementia (13,517 cases and 325,306 controls) and VaD (2048 cases and 328,982 controls) were obtained from the FinnGen consortium [14]. Summarized data for AD were obtained from the International Genomics of Alzheimer’s Project (IGAP), including 21,982 cases and 41,944 controls from 46 case–control studies [15]. Detailed information on GWAS datasets is summarized in Appendix A. Furthermore, we accessed the potential bias attributable to participant overlap utilizing a web-based tool (https://sb452.shinyapps.io/overlap/, accessed on 2 July 2023) and estimated type 1 error rates < 0.05 [16].

### 2.3. Selection of IVs

To obtain unbiased estimates of the causal effects, MR analysis should adhere to three assumptions: relevance, independence, and exclusion restriction. First, we extracted SNPs at the conventional genome-wide significance level (*p* < 5 × 10^−8^) with a linkage disequilibrium (LD) threshold of *r*^2^ < 0.1 within 500 kilobases’ (kb) distance. If the number of SNPs for metabolites was fewer than 3, we adopted a relaxed genome-wide threshold (*p* < 1 × 10^−5^) in reference to previous studies [17,18]. Steiger filtering was used to remove SNPs that were correlated with outcomes stronger than exposures [19]. We then measured the explained genetic variation (*R^2^*) utilizing the MR Steiger directionality test. The *F* statistic was computed by the formula *F* = *R*^2^(*N* − *K* − 1)/*K*(1 − *R*^2^), where *N* is the sample size of the GWAS for each metabolite and *K* is the number of IVs [20]. Metabolites with an *F* statistic of less than 10 were excluded to avoid the potential weak instrument bias [21]. Finally, the metabolites with more than 2 SNPs were included in the MR analysis [22]. Detailed information on IVs is presented in Appendix A.

### 2.4. Univariable MR

Inverse-variance weighting (IVW) was adopted to assess the causal effects of metabolites on outcomes (Appendix A). To render the conclusions more reliable, we conducted several sensitivity analyses for the identified significant metabolites (Appendix A). First, five MR models were utilized as complementary methods to evaluate the consistency and robustness of the results, including MR-Egger, weighted median, penalized weighted median, maximum likelihood methods, and MR pleiotropy residual sum and outlier (MR–PRESSO) [23,24]. Then, the Cochran *Q* test and MR-Egger intercept test were carried out to establish the existence of heterogeneity and horizontal pleiotropy, respectively. The MR-PRESSO global test was further applied to detect horizontal pleiotropy and possible outliers. Moreover, we performed the MR Steiger directionality test to validate whether the observed causalities were biased owing to reverse causation [19]. Finally, we also employed the PhenoScanner function in the MendelianRandomization package to exclude SNPs that were associated with education, body mass index, blood pressure, smoking and drinking at the threshold of *p* < 1 × 10^−5^ because these factors are known to be primary contributors to mortality and are also significant risk factors for outcomes [25]. The IVW was repeated after dropping the above SNPs to ensure robustness.

A reliable and robust causal association between circulating metabolites and dementia was determined in compliance with the following criteria: (1) the IVW method demonstrated a significant difference (*p* < 0.05); (2) consistent direction and magnitude among the five complementary MR methods; (3) no heterogeneity or pleiotropy was detected by Cochran’s Q test, MR-Egger intercept test, or MR-PRESSO global test (*p* > 0.05); (4) the MR-Steiger directionality test indicated that the effect direction from metabolites to dementia was true; and (5) the MR estimates remained significant after excluding SNPs associated with potential confounders.

### 2.5. MR-BMA Analysis

MR-BMA, an innovative expansion of multivariable MR using the Bayesian framework, shows the advantage of selecting and prioritizing highly correlated candidate risk factors in high-dimensional datasets [26]. MR-BMA analysis was further conducted in the subcategory showing significant enrichment of metabolic traits. We pooled SNPs that were associated with all selected lipid-related traits (*p* < 5 × 10^−9^) and strictly clumped them with an LD threshold of *r*^2^ < 0.001 within 10,000 kb (Appendix A). Posterior probability (PP) was calculated for each specific model to prioritize the best model. After the initial analysis, we calculated Cochran’s *Q* statistics and Cook’s distance for the models with PP > 0.02 to screen potential outliers. Subsequently, MR-BMA analysis was repeated after excluding the outliers (Appendix A). The marginal inclusion probability (MIP), representing the sum of the PP, was used to prioritize the candidate risk factors. The model-averaged causal effects (MACE) demonstrated the average causal effect of each lipid-related trait on the outcomes. Empirical *p* values for the MIP of metabolites were computed with 1000 permutations [27].

### 2.6. Statistical Analysis

All MR estimates are presented as odds ratios (ORs) with 95% confidence intervals (CIs) of outcomes. We adopted a conservative Bonferroni-adjusted threshold of *p* < 1.03 × 10^−4^ (0.05/486) for primary analyses and *p* < 4.07 × 10^−4^ (0.05/123) for secondary analyses to determine a statistically significant causal relationship. The metabolites with a two-sided *p* < 0.05 but above the Bonferroni-adjusted threshold were considered potential risk predictors for dementia. All analyses were conducted using R software (version 4.3.1) with the R packages TwoSampleMR, MendelianRandomization, and MR-PRESSO. The R-code for MR-BMA was accessible on GitHub (https://github.com/verena-zuber/demo_AMD, accessed on 15 March 2023).

### 2.7. Metabolic Pathway Analysis

The metabolites identified by IVW (*p* < 0.05) were imported into the pathway analysis module in MetaboAnalyst 5.0 (https://www.metaboanalyst.ca/, accessed on 5 May 2023). Metabolic pathway analysis was conducted utilizing a hypergeometric test. We tested human metabolic pathways from two metabolite set libraries: the Kyoto Encyclopedia of Genes and Genomes (KEGG) and the Small Molecule Pathway Database (SMPDB).

## 3. Results

### 3.1. Strength of the IVs

The analysis included a total of 472 metabolites for dementia and VaD, 473 metabolites for AD analysis, and all 123 metabolites were retained for validation. The *F* statistics of metabolites ranged from 20 to 504, indicating a considerable strength of the genetic instruments employed.

### 3.2. Univariable MR Analyses

A total of 186 causal features, corresponding to 118 unique metabolites, were preliminarily identified using IVW (Figure 2 and Appendix A). After the Bonferroni adjustment, we detected 44 statistically significant causal features. A total of 68 causal features demonstrated consistent associations in the sensitivity analyses (Appendix A), with 11 of them reaching the Bonferroni-adjusted threshold (Figure 3). Specifically, the genetically determined concentrations of large HDL particles (OR = 0.90, 95% CI: 0.85–0.95, *p* = 2.41 × 10^−4^), very large HDL particles (OR = 0.89, 95% CI: 0.85–0.94, *p* = 5.05 × 10^−5^), and small HDL particles (OR = 1.18, 95% CI: 1.09–1.29, *p* = 9.69 × 10^−5^) were causally associated with dementia susceptibility. The subfractions of large HDL particles (total lipids, free cholesterol, and phospholipids) and very large HDL particles (total cholesterol, cholesterol esters, and phospholipids) were associated with decreased risks of dementia. Regarding the subtypes of dementia, epiandrosterone sulfate (OR = 0.60, 95% CI: 0.48–0.75, *p* = 4.74 × 10^−6^) and glycoproteins (OR = 0.89, 95% CI: 0.83–0.95, *p* = 3.06 × 10^−4^) were associated with decreased risk of AD and VaD, respectively. Additionally, we identified 46 potential risk predictors that remained robust in the sensitivity analyses, particularly amino acids and acylcarnitines (Figure 4). With regard to the potentially shared molecules, 3-dehydrocarnitine was simultaneously proven to be related to a lower risk of AD and VaD.

### 3.3. MR-BMA Analyses

We performed an MR–BMA analysis with lipid-related traits among 123 metabolites identified by the IVW (Table 1 and Table 2). For dementia, the top-ranked model included only the mean diameter for HDL particles, followed by the concentration of very large HDL particles, and these two metabolite traits were identified as the dominant risk factors for the risk of dementia with the highest MIP. For AD, the top-ranked model included only total cholesterol in very large HDL particles, and it was also the metabolite trait with the strongest overall evidence. For VaD, the top-ranked model included triglycerides in very large HDL particles, followed by omega-7, omega-9, and saturated fatty acids, and the latter metabolite trait of fatty acids had the highest MIP rank.

### 3.4. Metabolic Pathway Analysis

As shown in Table 3, a total of 17 significant metabolic pathways (hypergeometric test, *p* < 0.05) were identified in the metabolic pathway analysis. Two shared metabolic pathways were implicated in the development of dementia, AD and VaD, including the “aminoacyl-tRNA biosynthesis” pathway and the “valine, leucine and isoleucine biosynthesis” pathway.

## 4. Discussion

This comprehensive MR study identified 118 metabolites relevant to the risk of dementia, 59 of which were lipids, supporting the crucial role of lipids in dementia pathogenesis. Our study revealed causal relationships of nine traits of HDL particles on dementia, epiandrosterone sulfate on AD, and glycoproteins on VaD. Our MR-BMA results indicated that the mean diameter of HDL particles and concentration of very large HDL particles had a predominant influence on the risk of dementia. Moreover, we detected 17 significant metabolic pathways involved in dementia, of which 2 were shared metabolic pathways among different subtypes of dementia.

Observational studies have documented the protective role of HDL cholesterol (HDL-C) on the risk of dementia [28]. In contrast, several studies have indicated that elevated levels of HDL-C are related to a higher risk of dementia [29,30]. The inconsistency observed in these findings can be attributed to variations in sample size, study designs, unmeasured confounders, and reverse causation, emphasizing the importance of employing the MR method to disentangle such scenarios. Moreover, HDL particles display remarkable heterogeneity in terms of structure, density, size, and composition, providing them with diverse functionalities that are essential to numerous biological processes [31]. Previous MR studies also show inconclusive evidence [32,33], possibly because they did not adequately consider the heterogeneity and complexity of HDL particles. Our comprehensive MR study clarified this controversial issue by conducting a meticulous analysis of specific subfractions and found protective effects of subfractions of large and very large HDL particles against dementia. Moreover, our study identified the predominant role of the mean diameter of HDL particles and the concentration of very large HDL particles in dementia. The following may explain the mechanisms underlying the involvement of HDL particles in dementia pathophysiology. Recent studies have proposed that HDL particle size serves as a critical determinant of HDL cholesterol efflux capacity [34] and may play a central role in mitigating cognitive decline. Furthermore, the lipid constituents encapsulated within HDL, such as phospholipids, cholesteryl esters, and free cholesterol, have been acknowledged as potent modulators of HDL functionality encompassing cholesterol efflux, vasodilation, antioxidative, and anti-inflammatory properties [35]. These multifaceted functions may contribute significantly to the pathophysiology of dementia, indicating their potential therapeutic relevance. Our study reaffirmed the potential health benefits of HDL in protecting against dementia [36] and emphasized the role of the mean diameter and concentration of HDL particles.

Regarding dementia subtypes, our study found protective effects of epiandrosterone sulfate and glycoproteins against AD and VaD, respectively. Epiandrosterone sulfate is classified as a sulfated steroid that reflects the metabolism of androgens [37]. Similarly, prior research has proposed the protective role of epiandrosterone sulfate against AD, which may improve plaque formation, enhance cognitive performance and increase longevity [38,39]. Glycoproteins, markers of inflammation, consist of protein molecules covalently linked to carbohydrate chains or glycans [40]. The increased expression of P-glycoprotein is a temporary physiological compensatory response in blood–brain barrier impairment to discharge undesirable substances [41]. A recent observational study proposed a glycoprotein to be a valid diagnostic biomarker for VaD [42]. Our study provides further compelling evidence, confirming the potential diagnostic value and therapeutic targets associated with glycoproteins in VaD.

Our study identified various potential risk predictors for dementia, with particular emphasis on amino acids and acylcarnitines, which were corroborated by previous observational studies [43]. Amino acids play multifaceted roles as neuromodulators, neurotransmitters and regulators of energy metabolism. Our study is consistent with existing evidence indicating the protective effect of essential amino acids against dementia [44]. Acylcarnitines represent derivatives of carnitine during the fatty acid transport to mitochondria for subsequent β-oxidation and participate in energy metabolism and neuroprotection [45]. Several studies have reported the beneficial impact of acylcarnitines on dementia, recognizing them as predictive diagnostic biomarkers for dementia [43].

The metabolic pathway analysis identified two shared pathways involved in dementia, AD, and VaD. The “aminoacyl-tRNA biosynthesis” pathway plays a vital role in protein synthesis and immune regulation [46], and may be related to the critical hypotheses explaining the pathophysiology of dementia. For the “valine, leucine and isoleucine biosynthesis” pathways, known as branched-chain amino acid (BCAA) pathways, previous studies have recognized abnormal metabolism of BCAA as the characteristic alteration in the development of dementia [47]. Furthermore, recent studies have indicated that fatty acid metabolism appears to be an essential determinant in the onset of dementia [48]. Our study further suggested that the “oxidation of branched-chain fatty acids” pathway was associated with VaD, consistent with the MR-BMA results highlighting the predominant role of fatty acids in VaD.

This study represents a comprehensive and systematic examination of the causal associations between circulating metabolites and dementia. Our study possesses several notable strengths. First, by utilizing genomic and metabolomic data, we have provided novel insights into the potential mediators of dementia and its subtypes through this systematic MR analysis. Second, by leveraging multiple large-scale GWASs, we were able to establish a robust causal inference with high statistical power. Additionally, we employed MR-BMA analysis, which facilitated the prioritization of the predominant metabolites associated with the risk of dementia in high-dimensional datasets. However, our research has several limitations deserving attention. First, we conducted a GWAS of circulating metabolites, which may not directly reflect the abnormal brain function associated with dementia. Future studies should focus on cerebrospinal fluid or brain tissue to provide more precise insights into the association of metabolites with dementia. Second, our study was exclusively from European populations, limiting extrapolation to other populations. It is crucial for future research to access the validity and applicability of our findings across different ethnicities. Third, the included GWAS datasets were heterogeneous in terms of population, sample and metabolic detection technology, which may contribute to the differences in metabolites identified by the primary and secondary analyses. We did not perform a meta-analysis owing to the limited overlap of metabolites. Furthermore, the MR method employed in our study assumes a lifetime exposure, which may not accurately represent reality. This assumption restricts our ability to identify potential nonlinear correlations between circulating metabolites and dementia. Exploring alternative analytical approaches that account for nonlinear associations could provide valuable insights into the complex relationship of metabolites with dementia.

## 5. Conclusions

In this systematic MR analysis, we identified nine traits of HDL particles as potential causal mediators of dementia, among which the mean diameter of HDL particles and the concentration of very large HDL particles played predominant roles in reducing the risk of dementia. Regarding dementia subtypes, epiandrosterone sulfate and glycoproteins were associated with decreased risk of AD and VaD, respectively.

## Figures and Tables

**Figure 1 nutrients-16-02879-f001:**
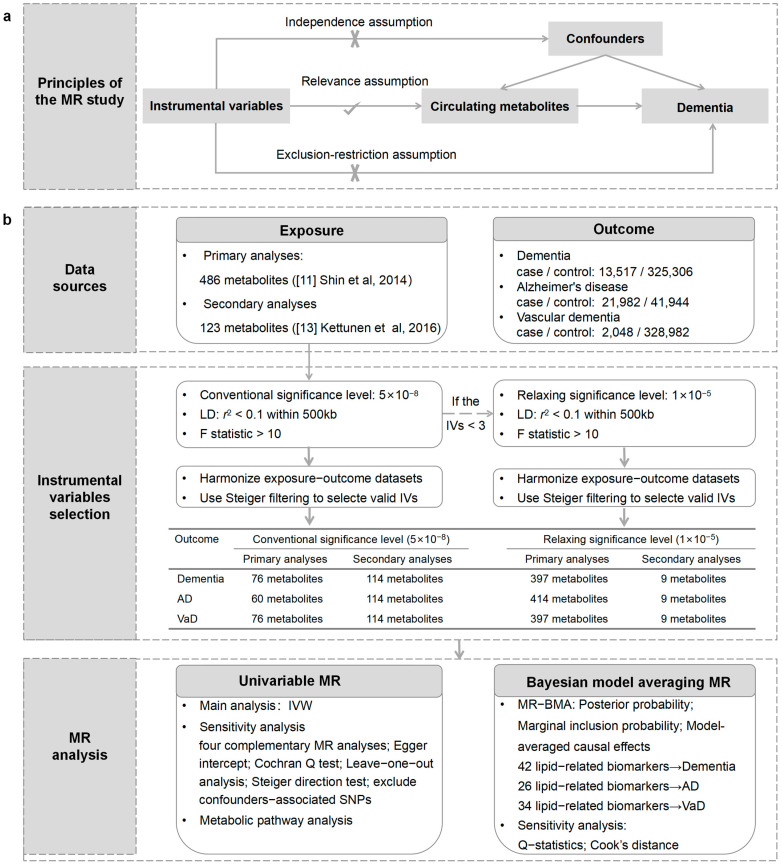
An overview of the study design. (**a**) The principles of the MR study. (**b**) Flowchart describing the sequence of analytical steps in this research. LD: linkage disequilibrium; IVs: instrumental variables; IVW: inverse-variance weighting; WM: weighted median.

**Figure 2 nutrients-16-02879-f002:**
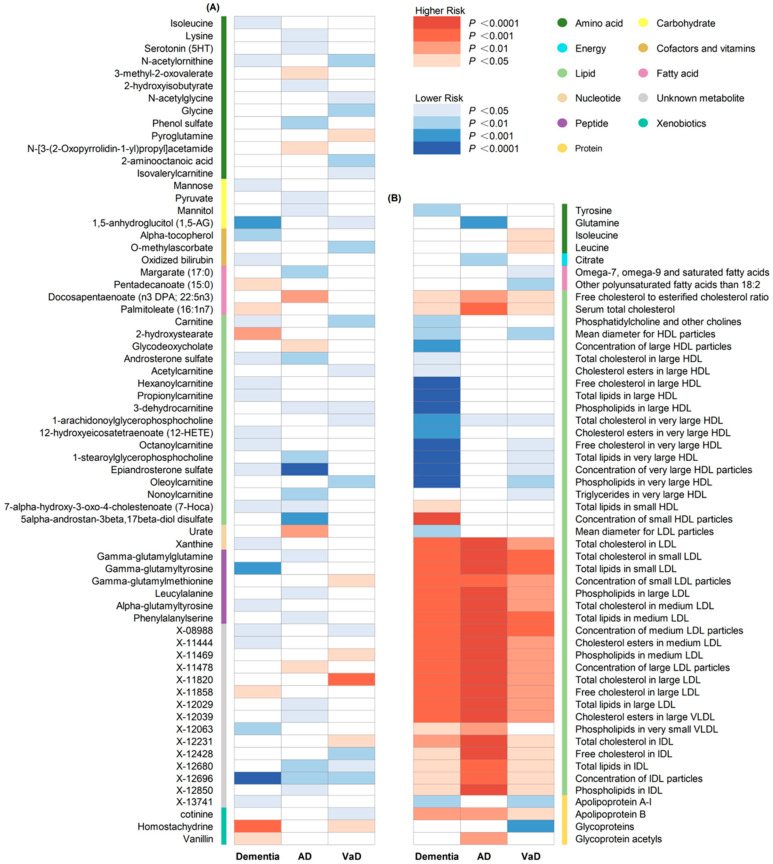
Heatmap illustrating the causal estimates of circulating metabolites on dementia, AD and VaD. (**A**) Heatmap showing the univariable MR analysis result of 486 circulating metabolites in the primary analyses; (**B**) heatmap showing the univariable MR analysis result of 123 circulating metabolites in the secondary analyses. AD: Alzheimer’s disease; VaD: vascular dementia; IVW: inverse-variance weighting.

**Figure 3 nutrients-16-02879-f003:**
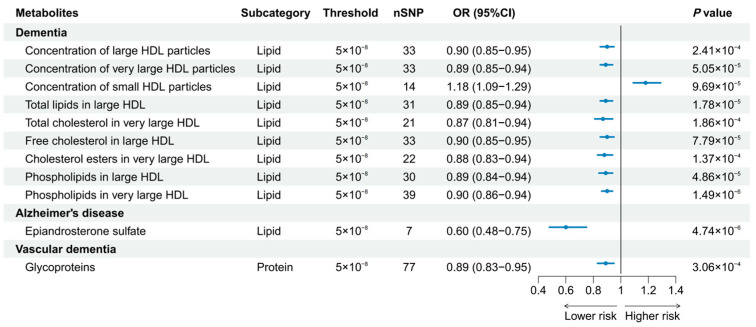
Forest plot of 11 causal features reaching the Bonferroni-adjusted threshold.

**Figure 4 nutrients-16-02879-f004:**
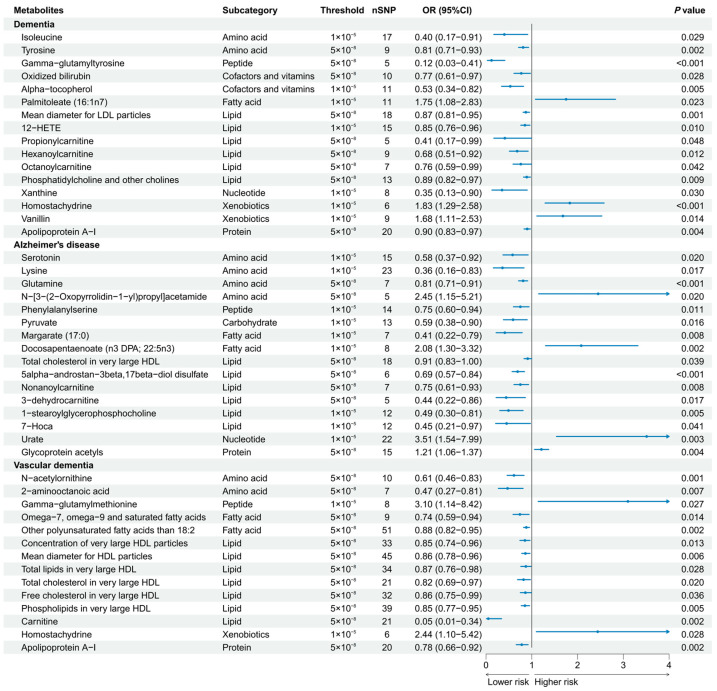
Forest plot of 46 potential risk predictors (*p* < 0.05) that remained robust in sensitivity analyses. 12-HETE, 12-hydroxyeicosatetraenoate; 7-Hoca, 7-alpha-hydroxy-3-oxo-4-cholestenoate.

**Table 1 nutrients-16-02879-t001:** Prioritization of the top 5 lipid-related traits for dementia, AD, and VaD.

Metabolite Traits	Rank	MIP	Average Effect	Empirical *p* Values
Dementia				
Concentration of very large HDL particles	1	0.139	−0.025	0.002
Mean diameter for HDL particles	2	0.127	−0.018	0.005
Free cholesterol in very large HDL particles	3	0.116	−0.02	0.097
Phospholipids in very large HDL particles	4	0.110	−0.015	0.028
Total lipids in very large HDL particles	5	0.084	−0.007	0.059
AD				
Total cholesterol in very large HDL particles	1	0.579	−0.094	0.005
Serum total cholesterol	2	0.101	−0.031	0.084
Free cholesterol to esterified cholesterol ratio	3	0.062	0.012	0.291
Glycoprotein acetyls	4	0.044	0.005	0.628
Phospholipids in medium LDL particles	5	0.044	0.007	0.985
VaD				
Omega-7, omega-9 and saturated fatty acids	1	0.311	−0.13	0.012
Serum total cholesterol	2	0.186	−0.103	0.001
Triglycerides in very large HDL particles	3	0.101	−0.02	0.026
Total cholesterol in medium LDL particles	4	0.101	0.039	0.012
Total cholesterol in LDL particles	5	0.1	0.041	0.007

MIP: Marginal inclusion probability; AD: Alzheimer’s disease; VaD: vascular dementia.

**Table 2 nutrients-16-02879-t002:** Models of lipid-related traits for dementia, AD, and VaD.

Model	Posterior Probability	Causal Estimate
Dementia		
Mean diameter for HDL particles	0.055	−0.109
Concentration of very large HDL particles	0.054	−0.124
Phospholipids in very large HDL particles	0.047	−0.112
Free cholesterol in very large HDL particles	0.042	−0.126
Total lipids in very large HDL particles	0.029	−0.115
Cholesterol esters in large HDL particles	0.028	−0.113
Concentration of large HDL particles	0.027	−0.112
Total lipids in large HDL particles	0.026	−0.112
Total cholesterol in large HDL particles	0.025	−0.114
Free cholesterol in large HDL particles	0.023	−0.114
Concentration of small HDL particles	0.020	0.161
AD		
Total cholesterol in very large HDL particles	0.362	−0.161
VaD		
Triglycerides in very large HDL particles	0.028	−0.170
Omega-7, omega-9 and saturated fatty acids	0.025	−0.246

AD: Alzheimer’s disease; VaD: vascular dementia.

**Table 3 nutrients-16-02879-t003:** Significant metabolic pathways (hypergeometric test, *p* < 0.05) involved in dementia, AD, and VaD.

Metabolic Pathway	Outcome	Database	Metabolites Involved	*p* Value
Aminoacyl-tRNA biosynthesis	Dementia	KEGG	Isoleucine, tyrosine	8.85 × 10^−3^
Aminoacyl-tRNA biosynthesis	AD	KEGG	Glutamine, lysine	2.33 × 10^−2^
Aminoacyl-tRNA biosynthesis	VaD	KEGG	Glycine, isoleucine, leucine	1.09 × 10^−4^
Valine, leucine and isoleucine biosynthesis	Dementia	KEGG, SMPDB	Isoleucine	2.56 × 10^−2^
Valine, leucine and isoleucine biosynthesis	AD	KEGG, SMPDB	3-methyl-2-oxopentanoic acid	4.06 × 10^−2^
Valine, leucine and isoleucine biosynthesis	VaD	KEGG, SMPDB	Leucine, isoleucine	1.39 × 10^−4^
Oxidation of branched chain fatty acids	Dementia	SMPDB	Carnitine, propionylcarnitine	6.46 × 10^−3^
Oxidation of branched chain fatty acids	VaD	SMPDB	Carnitine, acetylcarnitine	4.36 × 10^−3^
Phenylalanine, tyrosine and tryptophan biosynthesis	Dementia	KEGG, SMPDB	Tyrosine	1.29 × 10^−2^
Ubiquinone and other terpenoid-quinone biosynthesis	Dementia	KEGG, SMPDB	Tyrosine	2.87 × 10^−2^
Phenylalanine metabolism	Dementia	KEGG, SMPDB	Tyrosine	3.19 × 10^−2^
Arginine biosynthesis	Dementia	KEGG	Tyrosine	4.44 × 10^−2^
Glyoxylate and dicarboxylate metabolism	AD	KEGG	Citrate, pyruvate, glutamine	4.00 × 10^−4^
Citrate cycle (TCA cycle)	AD	KEGG, SMPDB	Pyruvate, citrate	4.20 × 10^−3^
Transfer of acetyl groups into mitochondria	AD	SMPDB	Pyruvate, citrate	1.01 × 10^−2^
Purine metabolism	AD	KEGG, SMPDB	Glutamine, urate	4.12 × 10^−2^
D-Glutamine and D-glutamate metabolism	AD	KEGG, SMPDB	Glutamine	3.06 × 10^−2^
Nitrogen metabolism	AD	KEGG, SMPDB	Glutamine	3.06 × 10^−2^
Valine, leucine and isoleucine degradation	VaD	KEGG, SMPDB	Leucine, isoleucine	3.77 × 10^−3^
Beta oxidation of very-long-chain fatty acids	VaD	SMPDB	Carnitine, acetylcarnitine	1.50 × 10^−3^
Carnitine synthesis	VaD	SMPDB	Carnitine, glycine	2.29 × 10^−3^
Arginine biosynthesis	VaD	KEGG	*N*-acetylornithine	3.57 × 10^−2^

AD: Alzheimer’s disease; VaD: vascular dementia; KEGG: Kyoto Encyclopedia of Genes and Genomes; SMPDB: Small Molecule Pathway Database.

## Data Availability

All data used are accessible at https://gwas.mrcieu.ac.uk/ (accessed on 3 April 2023), www.finngen.fi/en (accessed on 20 April 2023), and http://www.metabolomix.com/the-metabolomics-gwas-server-gwas-eu/ (accessed on 2 April 2023).

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
