# Peer review of "Causal Association between Circulating Metabolites and Dementia: A Mendelian Randomization Study"

_nutrients, 2024, doi:10.3390/nu16172879_

Round 1

Reviewer 1 Report

Comments and Suggestions for Authors

The authors conducted a two-sample Mendelian randomization analysis to examine the association of metabolites with risk of dementia, Alzheimer’s disease, and vascular dementia. They found 118 metabolites with evidence to support a causal association with risk of dementia and 17 significant metabolic pathways. The paper is overall very good quality and I have just a few minor comments: 

1) In Figure 1, in the instrumental variables selection section, it is not clear what is meant by the two numbers of metabolites for each outcome, e.g. dementia: 76 metabolites; 114 metabolites. 

2) Figure 2 shows that there were many more significant associations in the analysis of the Kettunen et al summary statistics than with the Shin et al summary statistics. The authors should include a discussion of the reasons for these differences. 

3) It is not clear what the difference is between Figures 3 and 4 from reading the captions. More precise and detailed captions are needed. 

4) There are no detailed methods describing how the pathways analysis was conducted. It just says that the MetaboAnalyst website was used but that is not sufficient to explain what was actually analysed. Furthermore Table 3 lists P-values for the ‘significant’ metabolic pathways but it is not at all clear what statistical test these P-values relate to.

Author Response

Reviewer #1:

The authors conducted a two-sample Mendelian randomization analysis to examine the association of metabolites with risk of dementia, Alzheimer’s disease, and vascular dementia. They found 118 metabolites with evidence to support a causal association with risk of dementia and 17 significant metabolic pathways. The paper is overall very good quality and I have just a few minor comments.

Response: Thank you for the positive comments. Our point-by-point responses to your comments are listed below.

Comments 1: In Figure 1, in the instrumental variables selection section, it is not clear what is meant by the two numbers of metabolites for each outcome, e.g. dementia: 76 metabolites; 114 metabolites. 

Response 1: Thank you for pointing this out. We agree with this comment and have made corresponding changes to Figure 1 to make it clear. We obtained two different sources of GWASs for circulating metabolites as exposures, corresponding to primary analyses (486 metabolites) and secondary analyses (123 metabolites), respectively. We extracted SNPs at the conventional (P <5×10−8) and relaxed (P < 1×10-5) genome-wide significance level to ensure that as many metabolites as possible could have appropriate instrumental variables (IVs). Therefore, at the conventional (P <5×10−8) level, we selected IVs for 76 metabolites in the primary analyses for dementia and 114 metabolites in the secondary analyses for dementia. Similarly, at the conventional (P <1×10−5) level, we selected IVs for 397 metabolites in the primary analyses for dementia and 9 metabolites in the secondary analyses for dementia.

Comments 2: Figure 2 shows that there were many more significant associations in the analysis of the Kettunen et al summary statistics than with the Shin et al summary statistics. The authors should include a discussion of the reasons for these differences.

Response 2: Thank you for your helpful comment. As suggested, we have provided a further discussion in the revised manuscript: “Third, the included GWAS datasets were heterogeneous in terms of population, sample and metabolomics detection technology, which may contribute to the differences in metabolites identified by the primary and secondary analyses. We did not perform a meta-analysis owing to the limited overlap of metabolites”.(Lines 336-340; Page 14).

Comments 3: It is not clear what the difference is between Figures 3 and 4 from reading the captions. More precise and detailed captions are needed. 

Response 3: Thank you for your insightful suggestion. We have provided more precise and detailed captions for these two figures in the revised manuscript. Figure 3 represents the forest plot of 11 causal features reaching the Bonferroni-adjusted threshold. Figure 4 represents the forest plot of 46 potential risk predictors (P < 0.05) that remained robust in sensitivity analyses.

Comments 4: There are no detailed methods describing how the pathways analysis was conducted. It just says that the MetaboAnalyst website was used but that is not sufficient to explain what was actually analysed. Furthermore Table 3 lists P-values for the ‘significant’ metabolic pathways but it is not at all clear what statistical test these P-values relate to.

Response 4: Thank you for your helpful comment. We have provided detailed methods describing how the pathways analysis was conducted in the revised manuscript: “The metabolites identified by IVW (P < 0.05) were imported into the pathway analysis module in MetaboAnalyst 5.0 (https://www.metaboanalyst.ca/). Metabolic pathway analysis was conducted utlizing a hypergeometric test.”.( Lines 239-240; Page 12) In addition, we have also described the detailed statistical tests corresponding to these P-values in Table 3: “Significant metabolic pathways (hypergeometric test, P < 0.05) involved in dementia, AD, and VaD.”.

Reviewer 2 Report

Comments and Suggestions for Authors

Neurodegenerative disorders appear as the main problem of further developed society. Most, of the studies presented in the literature concentrated on genetic information modification, DNA damage, etc. while the protective role of nutrition against Alzheimer's disease and dementia has been less attractive. This article entitled: Causal association between circulating metabolites and  dementia: A Mendelian randomization study have taken into consideration the physiological metabolites. The authors decide to concentrate on the European population. Presented studies have been correctly presented. Moreover, the Mendelian randomization methods have been used. Additionally, 17 metabolic pathways were identified which can play a significant role in the discussed disorder. However, the main point of this study is lipide role confirmation in AD and VaD. The above strictly indicates how a huge role plays in the correct nutritional profile in our lives.

The article is very well-written and readable with correctly selected references. I did not find points for language criticism. In my opinion, the article is valuable not only from the point of the topic but also from the point of available data analysis by open-source algorithms. All the above can be a good example for medical students’: haw to use open source for significant studies.

Author Response

Reviewer #2:

Neurodegenerative disorders appear as the main problem of further developed society. Most, of the studies presented in the literature concentrated on genetic information modification, DNA damage, etc. while the protective role of nutrition against Alzheimer's disease and dementia has been less attractive. This article entitled: Causal association between circulating metabolites and  dementia: A Mendelian randomization study have taken into consideration the physiological metabolites. The authors decide to concentrate on the European population. Presented studies have been correctly presented. Moreover, the Mendelian randomization methods have been used. Additionally, 17 metabolic pathways were identified which can play a significant role in the discussed disorder. However, the main point of this study is lipide role confirmation in AD and VaD. The above strictly indicates how a huge role plays in the correct nutritional profile in our lives.

The article is very well-written and readable with correctly selected references. I did not find points for language criticism. In my opinion, the article is valuable not only from the point of the topic but also from the point of available data analysis by open-source algorithms. All the above can be a good example for medical students’: haw to use open source for significant studies.

Response: Thank you for the positive comments.